# The Effect of Cylindrical Liner Material on the Jet Formation and Penetration Capability of Cylinder-Cone-Shaped Charge

**DOI:** 10.3390/ma15103511

**Published:** 2022-05-13

**Authors:** Yifan Wang, Zhijun Wang, Yongjie Xu, Zhen Jin

**Affiliations:** 1College of Mechanical and Electrical Engineering, North University of China, Taiyuan 030051, China; wyfnuc@163.com (Y.W.); yongqiang515@nuc.edu.cn (Y.X.); 2Chongqing Hongyu Precision Industrial, Chongqing 402760, China; 3Complex Systems Research Center, Shanxi University, Taiyuan 030006, China; jinzhn@263.net

**Keywords:** cylinder-cone-shaped charge, cylindrical liner, metallic material, numerical simulation

## Abstract

The jet formation and penetration capacity of cylinder-cone-shaped charges against steel targets were studied using the method of numerical simulation. Cylinder-cone-shaped charge models with five cylindrical liner materials, including nickel, tungsten, tantalum, steel 4340 and copper, were established to investigate the penetration capability and were compared with the classical conical-shaped charge. Moreover, the influence of the connection method of the cylindrical liner and the truncated liner on the jet performance was examined. The results show that the head velocity of the projectile formed by the cylinder-cone-shaped charge with a cylindrical nickel liner was larger than that with other cylindrical liner materials; in addition, it was larger by 50.2% compared with that formed by the classical conical-shaped charge. The penetration depth of the steel target by the cylinder-cone-shaped charge with a cylindrical copper liner was the largest, which could be 51.7% higher than that of a classical conical-shaped charge at a standoff of 2.5 D. For 2.0 D and 2.5 D standoff distances, the penetration depths were increased by 18.4% and 29.5%, respectively, by using the connection method of putting the cylindrical nickel liner on the neck of the truncated liner compared with that of the previous cylinder-cone-shaped charge with a cylindrical nickel liner.

## 1. Introduction

Shaped charges have been widely used in military and civilian fields as a technology to generate high energy density, especially for use against armored vehicles [1,2,3,4,5]. In recent years, the protection capabilities of the new armors have been continuously improved. When using the traditional-shaped charge, it is difficult to achieve efficient damage to the targets protected by high-strength armor. Therefore, the penetration performance of shaped charges must be enhanced to meet actual military needs [6,7,8]. It is crucial to study the structures and materials of the shaped charges and raise the performance.

In order to improve the velocity and penetration depth of shaped jets and the effective mass fraction of jets, many scholars have studied the structures and materials of shaped charges. Wang Cheng et al. [9] studied shaped charge with a truncated conical titanium alloy liner and an additional device. They showed that the structure could significantly improve the overall velocity of a shaped jet and form a super-shaped jet. Gu Wenbin et al. [10] studied the jet-forming process of a cylinder-cone liner, which mainly included changes in the jet head speed and the jet length with time. Xi et al. [11] comparatively studied the penetration ability of jets formed by W-, W–Ni–Fe-alloy- and W–Cu-alloy-shaped charge liners. It was found that W–Cu SCL, with the lowest density, exhibited the deepest penetration depth. Zhao et al. [12] studied the effects of Ni and Zn added to W–Cu alloy liners on the penetration mechanism and penetration performance of the shaped charge. The results show that Zn and Ni evidently reduced the penetration performance of the shaped charge liners. Hirsch et al. [13] studied the cohesion limit of a jet formed by a copper liner with a cone angle of 20° through experiments. Their results demonstrated that the cohesion limit of the jet was that the flow velocity was equal to 1.23 times the constant volume sound velocity of the material. Chanteret [14] conducted an experimental study on the limit value of jet cohesion proposed by Hirsch et al. (1.23 times the material constant volume sound velocity), and found that this criterion is valid for iron, molybdenum and nickel materials, but not for aluminum materials. The results of Kelly et al. [15] show that a jet will change from condensation to divergence when the flow reaches the critical Mach number of the material, and the experimental results also prove this conclusion. Xu et al. [16] examined the penetration capability of hypervelocity-shaped charge with silicon carbide, steel 45#, steel 4340, copper and tungsten discs using numerical simulation and experimental methods. Among the five materials, the hypervelocity-shaped charge with a tungsten disc formed a projectile with greater head velocity and length and produced a deeper penetration depth than other hypervelocity-shaped charges with other disc materials. In addition, the penetration depth of the classical conical-shaped charge was 379.1 mm at the standoff distance of 2.0 D (D: charge diameter). Based on the above research, it can be concluded that improving the structure of a liner improves the penetrating power of the shaped charge, and the selection of its material has gradually become a research hotspot.

Therefore, in this study, the jet formation process and penetration capability of cylinder-cone-shaped charges with five different cylindrical liner materials are researched. Firstly, we present the configurations of the cylinder-cone-shaped charge together with finite element models and material models of the cylinder-cone-shaped charges. Secondly, the effect of five different cylindrical liner materials in cylinder-cone-shaped charges is evaluated in terms of the shape, velocity of the projectile and depth of penetrated steel target using numerical simulation. Then, the effect of the connection method on the jet performance and penetration ability is studied. Finally, the research results are summarized.

## 2. Numerical Simulation

### 2.1. The Structure of a Cylinder-Cone-Shaped Charge

As shown in Figure 1, a cylinder-cone-shaped charge is composed of explosive material, a cylindrical liner and a truncated copper liner. The shell of the shaped charge is ignored in numerical simulation and filled with HMX. The charge diameter D is 100.0 mm and the charge length L is 120.0 mm. The thickness (Φ) of the cylinder-cone liner is 2.5 mm and the cone angle (α) of the truncated copper liner is 60°. Both the diameter (d) and length (l) of the cylindrical liner are 30.0 mm. In this research, the five materials of the cylindrical liner that are examined include nickel, tungsten, tantalum, steel 4340 and copper. Correspondingly, the classical conical-shaped charge has the same explosive and charge size, while the liner is a full cone.

### 2.2. Finite Element Model

In this study, overall simulations are performed using Autodyn [17]. Figure 2a shows the initial condition of the cylinder-cone-shaped charge model. The structure is axisymmetric; thus, only 1/2 of the model needs to be established. The nickel, tungsten, tantalum, steel 4340 and copper cylindrical liners are simulated using the Euler method. The computational area is 1000 mm × 200 mm, and the size of the mesh is set as 1 mm × 1 mm to preserve reasonable accuracy and computational costs. The air domain boundary type is defined as flow-out. Secondly, a steel target with a size of 600 mm × 100 mm is established using the Lagrange approach to study the penetration capacity of the projectile, as shown in Figure 2b.

### 2.3. Modeling Material

The equation of state (EOS) of air is ideal gas, which can be used in many applications involving the motion of gases. This equation can be derived from the laws of Boyle and Gay-Lussac, and it is expressed as
(1)$$P_{A}=\left(\gamma -1\right)\rho_{A}E_{A}$$
where $\rho_{A}$ is the density of air, which is 1.225 × 10^−3^ g/cm^3^; γ, as the gas constant, is 1.4, $E_{A}$ is 206.8 kJ/m^3^.

The EOS for the HMX explosive is “Jones–Wilkins–Lee” (JWL), i.e.,
(2)$$P_{E}=A\left(1-\frac{\omega }{R_{1}V}\right)e^{-R_{1}V}+B\left(1-\frac{\omega }{R_{2}V}\right)e^{-R_{2}V}+\frac{{\omega E}_{0}}{V}$$
where P_E_ is pressure, V = 1/ρ is relative volume, ρ is the density of HMX, $E_{0}$ is specific internal energy per unit mass of HMX, and all other terms are constants. The values of these parameters are given in Table 1.

The EOS of nickel, tungsten and tantalum is based on a shock model. The shock EOS describes the relationship between the pressure and internal energy at a point outside the Hugoniot curve; for the point on the curve,
(3)$$P=P_{H}+\Gamma \rho \left(E-E_{H}\right)$$
where $\Gamma_{\rho }=\Gamma_{0}\rho_{0}=constant$, ρ denotes the density, and Γ denotes the Grüneisen coefficient that is often approximated as 2. Furthermore, P_H_ and E_H_ represent the Hugoniot pressure and energy, respectively. The specific parameters of the material models were obtained from the material library in Autodyn [17]. The EOS parameters of nickel, tungsten and tantalum materials are listed in Table 2.

The Steinberg–Guinan model is used for the strength model of nickel, tungsten and tantalum. It puts the yield strength and shear modulus as functions of effective plastic strain, pressure, and internal energy (temperature).

As for steel 4340, the EOS is a linear model and the strength model is depicted by the Johnson–Cook equation, which defines the yield stress Y as
(4)$$Y=\left(A+{B\varepsilon }_{P}^{n}\right)\left(1+C\log\varepsilon_{P}^{*}\right)\left(1-T_{H}^{m}\right)$$
where $\varepsilon_{P}$ is the effective plastic strain. A, B, C, n and m are constants. ${Ε}_{P}^{*}=\varepsilon_{P}^{\cdot }/\varepsilon_{0}^{\cdot }$ with $\varepsilon_{0}^{\cdot }=1$ s^−1^ is the standardized effective plastic strain rate. $T_{H}^{m}$ is the homologous temperature, which is calculated as
(5)$$T_{H}^{m}=\left(T-T_{room}\right)/\left(T_{melt}-T_{room}\right)$$
where $T_{melt}$ indicates the melting temperature and $T_{room}$ indicates the room temperature. Parameters are listed in Table 3.

## 3. Results and Discussion

### 3.1. Comparison of Shaped Jet Formation

A representative formation of the projectile from a cylinder-cone-shaped charge is shown in Figure 3. The detonation wave begins to move towards the liner after the explosive is detonated. At 7 μs, the detonation wave acts on the cylindrical copper liner. The surface of the cylindrical liner is subjected to radial force and is then squeezed to the center. At 14 μs, the truncated liner begins to collapse and collides with the projectile from the cylindrical liner (see Figure 3d). Later, the liner continues to collapse, and the jet is stretched to form a high-speed projectile.

The impact impedance of the material can be approximated by R = ρc, where c is the sound speed of the material, so the impact impedance of copper is 3.53 × 10^7^ N∙s/m^3^, the impact impedance of nickel is 4.08 × 10^7^ N∙s/m^3^, the impact impedance of steel 4340 is 4.64 × 10^7^ N∙s/m^3^, the impact impedance of tantalum is 5.69 × 10^7^ N∙s/m^3^ and the impact impedance of tungsten is 7.77 × 10^7^ N∙s/m^3^.

Projectiles formed by cylinder-cone-shaped charges with five cylindrical liner materials (nickel, tungsten, tantalum, steel 4340 and copper) at 50 μs are shown in Figure 4. We can observe that the jet at the head of the projectile formed by the cylinder-cone-shaped charges with cylindrical nickel, steel 4340 and copper liners is formed from part of the cylindrical liner. However, the jet at the head of the projectile formed by the cylinder-cone-shaped charges with a cylindrical tungsten liner and cylindrical tantalum liner is formed from part of the truncated copper liner. This is because of two main influencing factors: the impact impedance of the material and the shape of the liner. When the impact impedance of different materials differs within a certain range, the shape of the liner plays a major role. The velocity of the jet formed by the partial cylindrical liner is greater than that of the truncated liner. Therefore, with the collapse of the truncated liner, the jet formed by the cylindrical liner is divided into two parts. The part with high velocity continues to move down the axis at the head of the jet, and the remaining part is located at the tail as a slug. However, when the impact impedance of different materials differs greatly, the impact impedance becomes the main influencing factor. The velocity of the jet formed by the material with high impact impedance is lower. The velocity of the jet formed by the cylindrical tungsten liner and the tantalum liner is lower than that formed by the truncated copper liner. So, the head of the jet is formed by the truncated copper liner. Among the five materials, cylinder-cone-shaped charges with a nickel and tungsten cylindrical liner generate the maximum and the minimum projectile length of 340.0 mm and 273.0 mm at 50 μs, respectively. The slug formed by the cylindrical tantalum liner accounts for a higher ratio of the entire length at 42.1%, and the total length of the projectile is 278.0 mm.

### 3.2. The Velocity of the Shaped Jet

In order to study the effect of the cylindrical liner materials on the jet velocity, eight fixed gauges are set on the basis of Figure 2a. The first gauge was set on the bottom of the shaped charge axis, with the next one set at a distance of 100 mm. The position of Gauge 3 is exactly the 2.0 D standoff distance. The variation curves of the jet velocity at these gauges are shown in Figure 5.

The velocities of the projectiles formed by cylinder-cone-shaped charges and the classical conical-shaped charge at the gauges are illustrated in Figure 5. Cylinder-cone-shaped charges can form jets with higher velocities compared to the classical conical-shaped charge. Among all five materials, the projectile formed by the cylinder-cone-shaped charge with a cylindrical nickel liner has the highest velocity, which is 8098.6 m/s at the 2.0 D standoff distance. In contrast, the velocity of the projectile formed by the classical conical-shaped charge is 5392.0 m/s at the standoff distance of 2.0 D. Cylinder-cone-shaped charges with cylindrical tantalum liners and tungsten liners yield nearly the same velocities of the projectiles at 6621.4 m/s and 6567.1 m/s at the 2.0 D standoff distance, respectively. In addition, the velocities of the projectiles formed by cylinder-cone-shaped charges with cylindrical copper liners and steel 4340 liners are roughly the same, measuring 7568 m/s and 7448.0 m/s at the 2.0 D standoff distance, respectively. We found that the jet velocity decreases with the increase in the impact impedance when the material of the cylindrical liner is different from that of the truncated liner. As for the cylinder-cone-shaped charges with cylindrical nickel, copper and steel 4340 liners, the reason for the increases in jet velocities is that the cylindrical liner can generate a high-velocity jet [18], so the head of the projectile is formed by the cylindrical liner. However, for high-impact impedance materials such as tantalum and tungsten, the increase in jet velocity is due to the fact that when the detonation wave is transmitted from the high-impedance medium to the low-impedance medium, the low-impedance material can obtain higher particle velocity, so the truncated copper liner obtains a higher velocity. Among the five materials, the fastest and slowest reductions in the jet head velocity are the cylinder-cone-shaped charges with cylindrical steel 4340 and nickel liners, respectively.

### 3.3. Penetration Depth

Figure 6 presents the penetration depths of all shaped charges in this work. Obviously, the cylinder-cone-shaped charges with cylindrical tantalum and copper liners show much longer penetration depths. For the 2.0 D standoff distance, the penetration depths of the cylinder-cone-shaped charges are greater than those of the classical conical-shaped charge. The penetration depth of the cylinder-cone-shaped charge with a cylindrical tantalum liner is greatest, measuring 468 mm, while the penetration depth of the classical conical-shaped charge is 335 mm, and hence, the penetration depth is increased by 39.7%. However, for the 2.5 D standoff distance, the penetration depths of the cylinder-cone-shaped charges with cylindrical nickel and tungsten liners are roughly the same, and are smaller than that of the classical conical-shaped charge. The cylinder-cone-shaped charge with a cylindrical copper liner yield the greatest penetration depth (472 mm), which is an increase of 51.7% compared to the penetration depth of the classical conical-shaped charge. The standoff distance has little effect on the penetration depth of the cylinder-cone-shaped charge with a cylindrical steel 4340 liner. The value with the 2.0 D standoff distance is slightly bigger than that with the 2.5 D standoff distance by 0.8%. What is more noteworthy is that the penetration depth of the cylinder-cone-shaped charge with a cylindrical copper liner is proportional to the standoff distance. However, the others show an opposite tendency.

The penetration progresses of the cylinder-cone-shaped charges and the classical conical-shaped charge at 2.0 D and 2.5 D standoff distances are presented in Figure 7. For the 2.0 D standoff distance, the entrance of penetration channels formed by cylinder-cone-shaped charges with cylindrical nickel, tungsten, tantalum and steel 4340 liners are funnel-shaped, but the entrance caused by a cylinder-cone-shaped charge with the cylindrical copper liner is cylindrical. However, for the standoff distance of 2.5 D, most entrances of penetration channels formed by cylinder-cone-shaped charges are cylindrical except those formed by the cylinder-cone-shaped charge with the cylindrical tantalum liner. During the process of penetration, the projectile breaks at the junction of the jet and the slug due to the large velocity difference. The jet with high velocity and small diameter plays a crucial role in the penetration depth of the target. For cylinder-cone-shaped charges with cylindrical nickel and steel 4340 liners, although the head velocity of the jet is formed by the cylindrical liner before it touches the target, as the jet penetrates the target, the jet formed by the truncated copper liner plays a major role in the penetration depth. After 150 μs, the penetration rates are significantly reduced. The entrance diameter of the target is mainly generated by the slug, and the diameter of the slug is larger than that of the jet.

### 3.4. The Influence of the Connection Method

For the cylinder-cone liner, how to combine the cylindrical liner and the truncated liner is a problem that needs to be considered. As shown in Figure 8, we extend the truncated liner, which is similar to the neck; its length is 20.0 mm and the thickness is 1.5 mm. The cylindrical liner is similar to a hat that is placed on the neck of the truncated liner. The material of the cylindrical liner is nickel, and the truncated liner is still copper. The effect of this connection method on the jet performance and penetration ability was studied using numerical simulation. Furthermore, a comparative analysis was conducted between the cylinder-cone-shaped charge with this connection method and the previous cylinder-cone-shaped charge.

The shape and velocity distribution of a projectile formed by cylinder-cone-shaped charge with the cylindrical nickel liner considering the connection method at 50 μs is shown in Figure 9a. Obviously, the continuity of the projectile head is poor, which is just a small mass of high-speed fragments, and this part has little ability to penetrate the target. However, the head velocity of the projectile is higher than that of the previous cylinder-cone-shaped charge. Figure 9b,c present the penetration depth of the cylinder-cone-shaped charge with the cylindrical nickel liner considering the connection method for 2.0 D and 2.5 D standoff distances. The penetration depths are greater compared to the previous cylinder-cone-shaped charge with the cylindrical nickel liner, and show an increase of 18.4% and 29.5% for 2.0 D and 2.5 D standoff distances, respectively. Therefore, this connection method can improve the penetration ability of the projectile, and the length and thickness of the neck of the truncated liner also affect the velocity and penetration ability of the projectile.

## 4. Conclusions

We researched the effect of five cylindrical liner materials, including nickel, tungsten, tantalum, steel 4340 and copper, on the jet formation of cylinder-cone-shaped charge and its penetration into steel targets using numerical methods in the present paper. In addition, a comparative analysis between a classical conical-shaped charge and cylinder-cone-shaped charges was performed. At the same time, the influence of the connection method of the cylindrical liner and the truncated liner on the jet performance was studied. The major findings are generalized as follows:(1)Among the five materials, the head jet formed by the cylinder-cone shaped charge with cylindrical nickel, steel 4340 and copper liner was formed from the part of cylindrical liner and that formed by the cylinder-cone shaped charge with cylindrical tungsten and tantalum liner was formed from the truncated copper liner.(2)The head velocity of the jet formed by the cylinder-cone-shaped charge with cylindrical nickel liner was largest among all of the tested materials. It measured 8098.6 m/s at the 2.0 D standoff distance, and the head velocity was increased by 50.2% compared with that of the jet formed by the classical conical-shaped charge.(3)The penetration depths of all cylinder-cone-shaped charges were greater than that of the classical conical-shaped charge for the 2.0 D standoff distance. The penetration depth of the steel target by the cylinder-cone-shaped charge with a cylindrical copper liner was largest, and could be 51.7% higher than that of a classical conical-shaped charge at a standoff of 2.5 D.(4)For 2.0 D and 2.5 D standoff distances, the penetration depths were increased by 18.4% and 29.5%, respectively, by using the connection method of putting the cylindrical nickel liner on the neck of the truncated liner.

## Figures and Tables

**Figure 1 materials-15-03511-f001:**
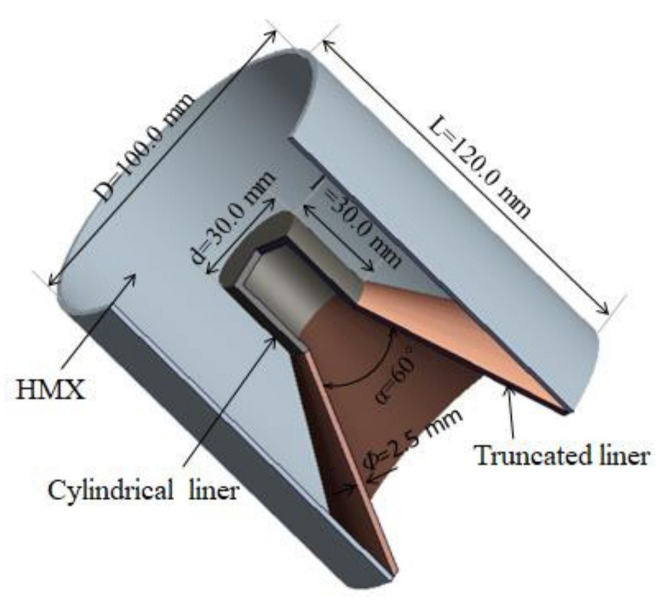
The structure parameters of the cylinder-cone-shaped charge.

**Figure 2 materials-15-03511-f002:**
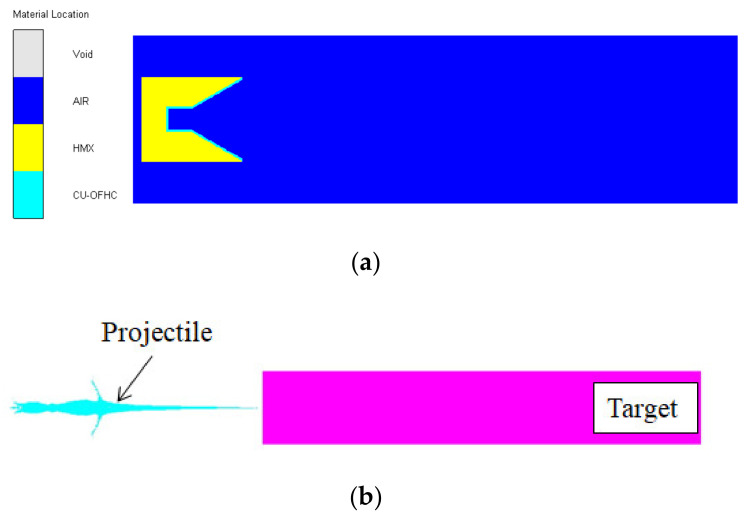
The numerical simulation model: (**a**) the initial condition of the cylinder-cone-shaped charge model; (**b**) the model of projectile penetrating steel target.

**Figure 3 materials-15-03511-f003:**
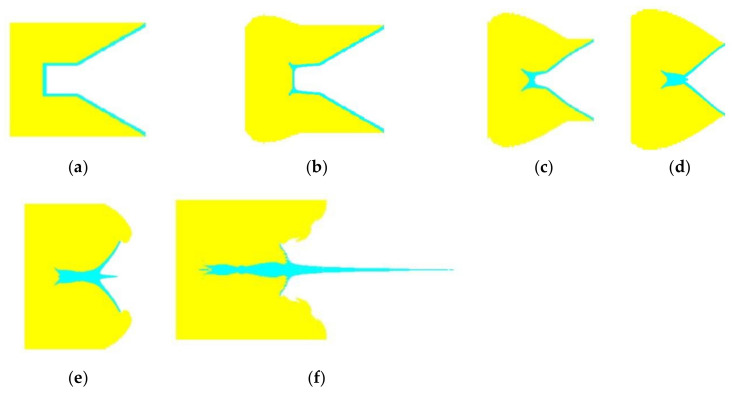
The formation process of the projectile formed by the cylinder-cone-shaped charge with a cylindrical copper liner. (**a**) *t* = 0 μs. (**b**) *t* = 7 μs. (**c**) *t* = 11 μs. (**d**) *t* = 14 μs. (**e**) *t* = 20 μs. (**f**) *t* = 55 μs.

**Figure 4 materials-15-03511-f004:**
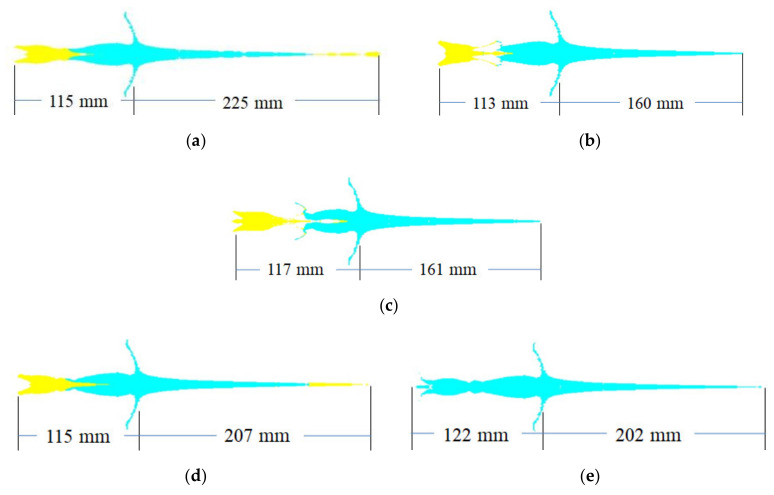
Projectiles of cylinder-cone-shaped charges at 50 μs. (**a**) Cylindrical nickel liner. (**b**) Cylindrical tungsten liner. (**c**) Cylindrical tantalum liner. (**d**) Cylindrical steel 4340 liner. (**e**) Cylindrical copper liner.

**Figure 5 materials-15-03511-f005:**
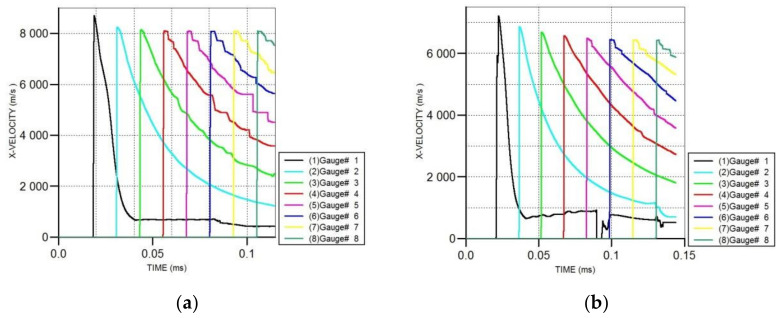
The variation curves of the jet velocity at gauges. (**a**) Cylindrical nickel liner. (**b**) Cylindrical tungsten liner. (**c**) Cylindrical tantalum liner. (**d**) Cylindrical steel 4340 liner. (**e**) Cylindrical copper liner. (**f**) Classical conical-shaped charge.

**Figure 6 materials-15-03511-f006:**
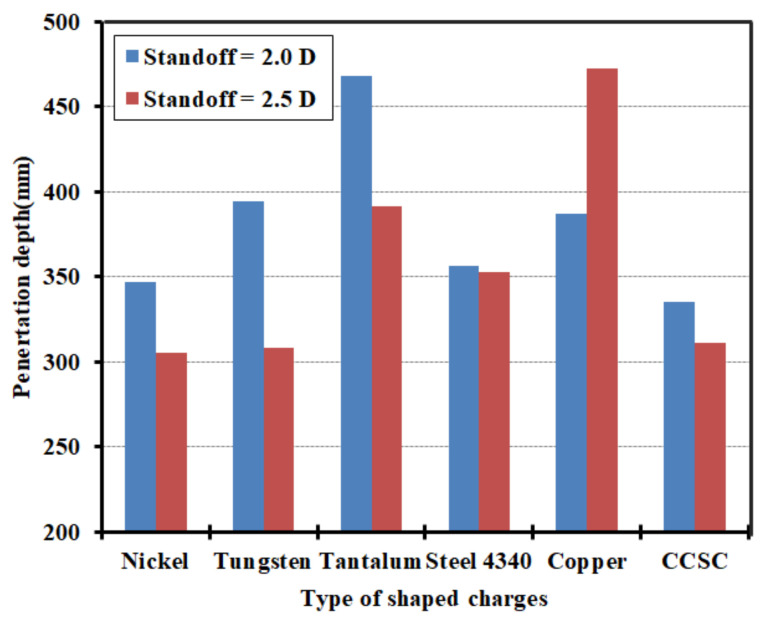
Penetration depths of shaped charges at 2.0 D and 2.5 D standoff distances (CCSC: classical conical-shaped charge).

**Figure 7 materials-15-03511-f007:**
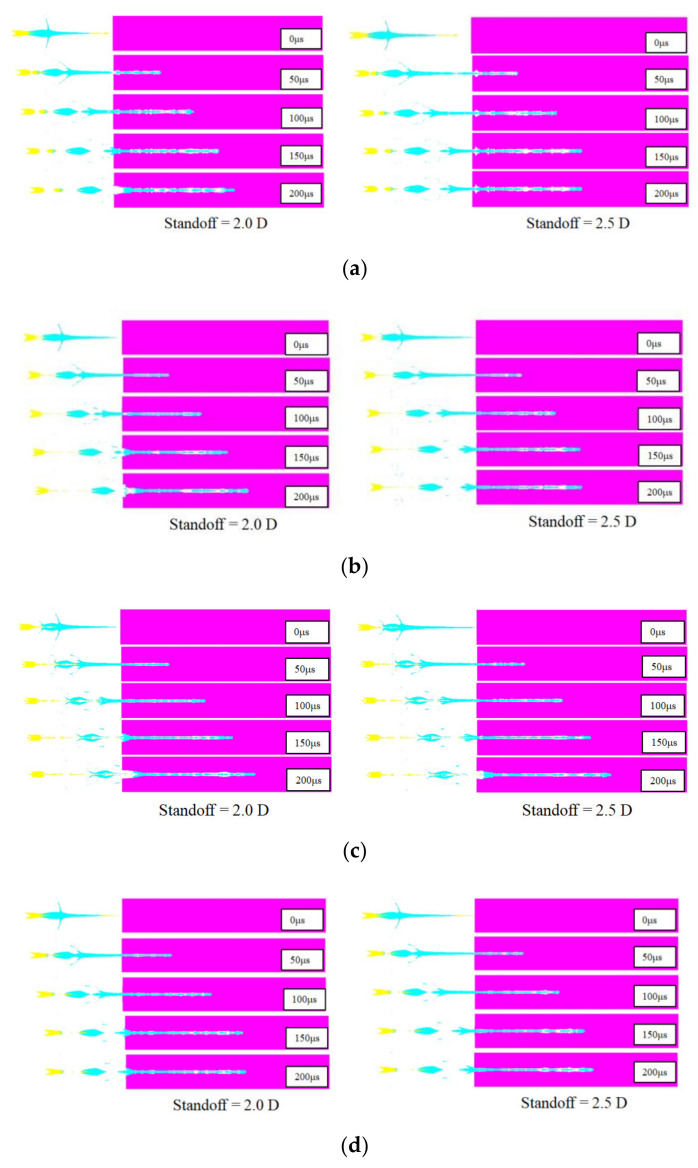
Penetration progresses of shaped charges. (**a**) Penetration progress of shaped charge with nickel cylindrical liner. (**b**) Penetration progress of shaped charge with tungsten cylindrical liner. (**c**) Penetration progress of shaped charge with tantalum cylindrical liner. (**d**) Penetration progress of shaped charge with steel 4340 cylindrical liner. (**e**) Penetration progress of shaped charge with copper cylindrical liner. (**f**) Penetration progress of classical conical-shaped charge.

**Figure 8 materials-15-03511-f008:**
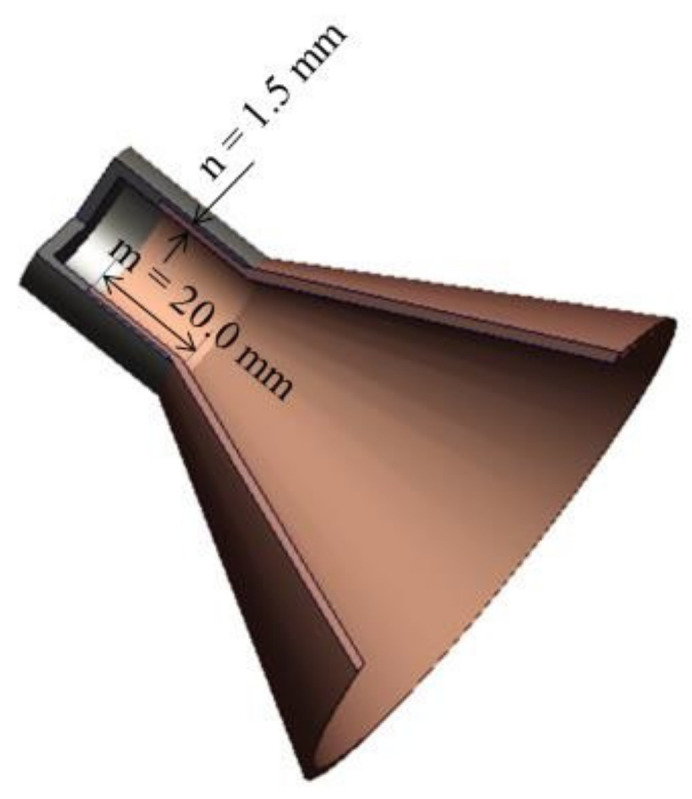
The connection method of the cylindrical liner and the truncated liner.

**Figure 9 materials-15-03511-f009:**
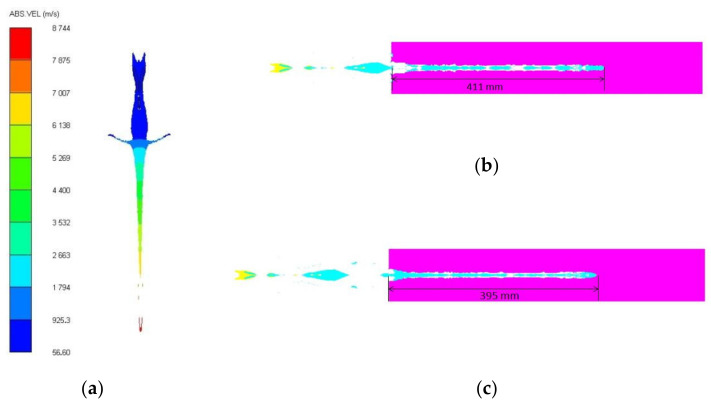
The velocity distribution of projectile and penetration depth. (**a**) The shape and velocity distribution of projectile at 50 μs; (**b**) penetration depth at the standoff distance of 2.0 D; (**c**) penetration depth at the standoff distance of 2.5 D.

**Table 1 materials-15-03511-t001:** The parameters in the JWL EOS of HMX.

ρ (g/cm^3^)	E_0_ (kJ∙m^−3^)	A (GPa)	B (GPa)	R_1_	R_2_	ω
1.891	1.05 × 10^7^	778.28	7.071	4.20	1.00	0.30

**Table 2 materials-15-03511-t002:** The EOS parameters of nickel, tungsten and tantalum.

Parameter	Nickel	Tungsten	Tantalum
EOS	Shock	Shock	Shock
ρ_0_ (g/cm^3^)	8.874	19.3	16.69
Γ (none)	1.93	1.67	1.67
C_0_ (m/s)	4602.0	4030.0	3410.0
S (none)	1.437	1.237	1.20

**Table 3 materials-15-03511-t003:** Parameters in the strength model of steel 4340.

*A* (Mpa)	*B* (Mpa)	*n*	*C*	*m*	*T_melt_* (K)	*T_room_* (K)
760.0	507.0	0.28	0.064	1.06	1793.0	300.0

## Data Availability

The data used to support the findings of this study are included within the article.

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
