# Peer review of "The Effect of Cylindrical Liner Material on the Jet Formation and Penetration Capability of Cylinder-Cone-Shaped Charge"

_materials, 2022, doi:10.3390/ma15103511_

Round 1

Reviewer 1 Report

The work performed is very important in terms of findings on 5 different materials shaped charges. I recognize the tremendous work performed to model and to perform accurate simulations. However, deeper explanations on the CFD model used were not provided and the simulations were not validated by real experiments.

Please perform a thorough check of English and acronyms designations. Also, some more references on previous works performed in the same area, using the same shaped charges, should be provided.

Reviewer 2 Report

The work is interestig and includes some new achievements. The value of the research would be higher in the Authors would present also at least one laboratory test presented the similar situation as shown in numerical tests. In computations as we know it is possible to generate any results but the fundamental question is how those results fit the real behaviour of the materials.

It would be also good to comment the constitutive relation used in computations for materials. Why the Johnson-Cook model is correct? Do the parameters of this model for the presented velocities of deformations correstond to the reality?

Small comparison with the experiments would clarify the value of the computations.

Reviewer 3 Report

Dear Authors,

The authors are well versed in the subject area, and in their article
reflected the current state of affairs.

Numerical simulation does not always imply a comparison with experiment;
it is enough to get into the confidence intervals of the key parameters
of the numerical calculation.

The use of copper and steel 4330 has new results. The most common and
studied material as a projectile is tungsten. Tungsten, tantalum and
copper have a significant density, in addition, tungsten and tantalum
increase the exothermic effect due to interaction with oxygen, and as a
result increase penetrating effect. Copper has the lowest melting point
of all the above materials, it passes into the state of gas more easily,
and favorable conditions are created for the formation of a cumulative
jet.

In my opinion, the use of tungsten, tantalum and copper is justified,
but for nickel, I would like to hear detailed author's explanations.
